# Factors That Drive Actual Purchasing of Groceries through E-Commerce Platforms during COVID-19 in Indonesia

**Dezie Leonarda Warganegara [1,*] and Roozbeh Babolian Hendijani [2]**

1 Management Department, BINUS Business School Doctor of Research in Management, Bina Nusantara University, Jakarta 11480, Indonesia
2 Creative Marketing Program, Management Department, BINUS Business School Master Program, Bina Nusantara University, Jakarta 11480, Indonesia; rhendijani@binus.edu
* Correspondence: dezie@binus.edu

**Abstract:** (1) Background: This is one of the few studies to look into online grocery shopping behavior in Indonesia, as an emerging sector of the economy. The technology acceptance model is extended in this study to include price, health risk, and a reference group to better understand the factors and the extent to which they influence online grocery shopping. (2) Methods: In order to achieve the goal of the research and test the research model, a literature-based questionnaire was developed and distributed to 300 respondents in Jakarta via online platforms. Partial Least Squares Structural Equation Modeling (PLS-SEM) was used in this study. (3) Results: We discovered that ease of use, usefulness, attitude, and reference group had a statistically significant relationship with intention and actual use of online platforms to purchase groceries in Indonesia. However, neither health risk nor price were found to be significantly correlated with respondents' purchasing intent. (4) Conclusions: Several practical and theoretical implications for decision makers designing marketing strategies are discussed based on the findings.

**Keywords:** online shopping; groceries; TAM; price; Indonesia

## 1. Introduction

COVID-19 is an ongoing worldwide pandemic that has affected human life and behavior all around the world. Indonesia is one of the countries that have been hit hardest by COVID-19. The government of Indonesia officially announced the COVID-19 outbreak in the middle of March 2020. According to official figures, Indonesia, the fourth most populous country in the world, has had more than 3 million coronavirus infections and more than 80,000 deaths [1] by the time of this study and is currently grappling with the worst coronavirus outbreak in Asia. Based on this situation, the government extended its restrictions on community activities (*PPKM darurat*) to prevent the spread of the virus [2]. There are many activities affected by this pandemic and one of them is online businesses. It was found that, by using e-commerce to satisfy the needs of the people, the government can achieve its goal of reducing the mobility of residents [3].

As in other countries, the pandemic is having an impact on consumer behavior in Indonesia [3]. People are concerned about the current situation and doubt whether they can remain virus-free [4]. Due to the government's social distancing regulations, which have resulted in changes in consumer behavior, there are limitations on movement and activities. Based on these regulations, many activities such as selling or purchasing products have moved to online platforms, and many customers use these platforms to meet their daily needs [5,6]. Online shopping is a rapidly evolving industry, as Internet technologies and applications provide customers with more accessible, convenient, and cost-effective ways to find a wider range of products than traditional shopping [7,8].

The growth of the Internet has already changed the lifestyles of consumers as well as their shopping behavior. Due to the Internet and mobile connectivity in Indonesia,

there is a high rate of e-commerce in Indonesia. The Indonesian Internet Service Providers Association (APJII) survey found that, of a total of 277 million people in Indonesia, there are 197.6 million (73.7%) Internet users [9]. Online shopping is among the reasons for the increase in Internet access, especially during the pandemic.

Online stores are viewed as a way for the community to fulfill needs without having to leave their homes. Online shopping has become more popular as a new lifestyle among Indonesians. The growing public interest in online shopping has influenced the rapid development of Indonesia's e-commerce industry. This is seen in the accelerated e-commerce growth in Indonesia, from 54% in 2019 to 91% in 2020 [10]. Online shopping platforms have enabled people to meet their needs while adhering to social distancing regulations. With the pandemic's restrictions on freedom of movement, online shopping has increased dramatically. During a crisis, consumers' preferences, such as what they buy, where they shop, and how frequently they shop, change [11]. A crisis such as COVID-19 also causes consumers to learn or adapt to new shopping habits [12]. Understanding consumer purchasing patterns during crises is critical to any business's success [13]. This trend is bolstered by the rise of e-commerce platforms in Indonesia. Each of those online shops offers a variety of attractive services and discounts on their platforms to increase their sales, while also helping the government to reduce the mobility of people.

Online-based economic growth influences changes in consumer behaviors, lifestyles, and activities. When shopping, consumers use the Internet as they value time, efficiency, and cost savings. They are also not required to visit physical stores and interact directly with sellers. Online shopping allows consumers to do their routine shopping through an online shop and make transactions directly on the same platform, which causes a surge in the migration of consumers to e-commerce to shop for groceries, and this surge is likely to continue. Unprecedented sales have been seen as consumers have started buying groceries online [14,15]. Even consumers who never bought groceries online in the past have been forced to switch to online shopping [16]. In the current study, "groceries" refers to fresh fruits and vegetables.

Despite the rapid increase in online shopping in Indonesia based on the above data [3,9], there is still limited research on the determinants of consumer attitudes and behavior related to online grocery shopping. A study by [17] shows that consumers in Hanoi, Vietnam, tended to buy groceries online and feel safe because of the low prices. However, the impact of the Technology Acceptance Model (TAM) and Price and Reference Group factor on consumer attitudes and intentions towards online purchases were not fully examined. Another limitation of previous studies is that they focused on final foods [17] or nonperishable products [6]; there is a lack of studies about customers purchasing perishable groceries through online platforms [16]. In addition, a comprehensive review [18] noted that TAM has been widely used in developed countries [18,19] and markets, but might not be applicable for growing economies. Furthermore, while the existing literature provides a comprehensive overview of various acceptance models and their explanatory power on purchasing groceries online in developed countries, such as the United States and in Europe, additional research is required to test their predictive ability in developing countries such as Indonesia. Therefore, more studies in countries with different cultures are needed than what is currently available in the literature.

The current study's main goal was to identify the factors influencing consumer attitudes and intentions regarding online grocery shopping in Indonesia's emerging economy to fill the current literature gap and gain more knowledge about online grocery shopping. Taking valuable lessons learned from previous research and analyses, this study expands TAM and adds other variables such as price [20,21], reference group [22,23], and health risk [24], which are crucial factors in online shopping behavior and predict the sustainable behavior of customers.

Sustainability is not a negotiable factor for online shops, especially in the COVID-19 era. In online shops, the concept of sustainability can include the business model, offers, and marketing strategies, and its role will become increasingly important in the coming

years. This was supported by a previous study [4] that found a link between perceived sustainability and customer engagement in e-commerce. The influence of e-commerce has changed people's lifestyle and has emerged alongside sustainable development. Therefore, its significance is considered to have an extraordinary impact on the modern world [12]. In order to be sustainable, online grocery shops must take into account e-commerce customers' unstable behavior.

The current study used a quantitative approach and primary data collected by distributing online questionnaire to 300 respondents. Partial least squares structural equation modeling (PLS-SEM) is a well-established method for analyzing complex causal relationships among latent variables. In the current research, this approach is used to validate the theoretical framework. As a result, it contributes to the ongoing discussion about the precursors and determinants of online grocery shopping. This study adds to existing knowledge about technology acceptance and online grocery shopping in developing and emerging countries by focusing on Indonesia. In practice, the results will provide new insights into how various factors are improving online food purchases in the developing Asian market of Indonesia. In essence, it would also help stakeholders, such as grocers, policy makers, and governments, in developing and managing strategies and initiatives to promote online grocery shopping.

The remainder of this research is organized as follows. Section 2 discusses and summarizes the relevant literature and formulates the hypothesis of this study. Section 3 details the data collection process and research methods applied in this study. Section 4 elaborates on the results and presents a discussion of the findings. The conclusions of the research, along with the theoretical and practical implications, limitations, and suggestions for future studies, are given in Section 5.

## 2. Literature Review

### 2.1. Technology Acceptance Model (TAM)

With the advancement of technology and media, online shopping has recently become a popular shopping method [25,26]. In recent years, there has been a steady increase in the number of online shoppers and online sales [27]. This was accelerated by the spread of COVID-19, which has pushed people to purchase more from online shops [5,6]. Online shopping has advantages over traditional shopping options because it is available anywhere and at any time [28]; it saves time [29]; it offers a wide range of products [30]; and it makes cost saving possible [31]. Previous studies based on the Technology Acceptance Model (TAM) developed by [32] prove that these advantages are among the most important positive influencing factors of intention [25,33,34]. In addition to the perception of usefulness, studies using TAM portray how the ease of use affects online consumers' purchase intention [34–36]. Online retailers should determine the factors that could hinder and promote online shopping intent to encourage consumers to purchase more [37]. To better understand the factors influencing the intent and actual behavior of Indonesian consumers when shopping for groceries online, this study will use TAM and add three factors, namely reference group, health risk, and price. By extending the TAM model and incorporating selected variables, this study plans to understand factors affecting the online purchasing behavior of customers in the COVID-19 era. By knowing more about the behavior of customers, companies can create appropriate marketing strategies. This study contributes to the literature by providing greater explanatory power for why Indonesian consumers prefer to shop for groceries online. Furthermore, this study investigates inconsistencies in previous studies.

### 2.2. Perceived Ease of Use

Ease of use is defined as "the degree to which an individual believes that using a particular system would be free of physical and mental effort" [32] (p. 985). In other words, perceived ease of use denotes perceptions concerning "the process leading to the final outcome" [38] (p. 104). The benefits of perceived ease of use in e-commerce include ease of

ordering at any time and from any location, perceived ease of information searching, and overall ease of use [38,39]. The authors of this research define perceived ease of use as a consumer's perception that online grocery shopping requires minimal effort.

TAM asserts that perceived usefulness is influenced by perceived ease of use. Previous research on online shopping has discovered that perceived ease of use influences perceived usefulness in both developed and emerging markets [39–41]. Therefore, consumers will perceive the usefulness of online shopping if it is easy to use by being connected to the Internet for purchasing products [42–45]. A study of online grocery shoppers reveals a significant and positive relationship between perceived ease of use and perceived usefulness [46]. Therefore, this study hypothesizes:

**Hypothesis 1 (H1).** *There is a significant and positive relationship between perceived ease of use and perceived usefulness of online grocery shopping.*

TAM [32] states that perceived ease of use and usefulness are both motivators of customer attitudes toward adopting a new technology or system. When customers perceive that Internet-connected gadgets or equipment are easy to use, a positive attitude is generated about online purchasing. Previous research discovered that customers' perceptions of the ease of use of retail websites and online shopping positively influenced their attitude to shop online. This was supported by a previous study conducted in an emerging economy that found that there is a significant and positive relationship between ease of use and attitude of customers [7]. Therefore, this study hypothesizes:

**Hypothesis 2 (H2).** *There is a significant and positive relationship between consumers' perceived ease of use and attitudes toward online grocery shopping.*

*2.3. Perceived Usefulness*

Based on [32] (p. 985), perceived usefulness refers to "the degree to which a person believes that using a particular system would enhance his/her job performance." In other words, perceived usefulness reflects individuals' perceptions concerning "the outcome of the experience" [38] (p. 104). Perceived usefulness of online shopping is related to the perceived benefits such as maximizing time savings and convenience. Applying this to the online grocery shopping context, the authors described perceived usefulness as consumers' perception that shopping for groceries online improves their shopping experience and performance. Moreover, online shopping can increase the efficiency of the entire online purchasing process, allowing consumers to compare prices from different retailers, search for product information, place an order, finalize the transaction, track the shipment, and assess the customer service [42,43,45]. Therefore, the perceived usefulness of online shopping can be conceptualized as the degree to which online shopping provides relative benefits to customers compared with offline shopping [40,45]. This was supported by previous studies about the positive relationships between perceived usefulness and customers' attitudes towards online shopping [43,44,46]. Perceived usefulness is also another major factor of attitude as it affects consumers' decision to shop online if they find online shopping to be useful [43,44]. This was supported by previous studies that found that the perceived usefulness of online shopping significantly and positively affects attitudes towards purchasing products from online shops [7,46].

Therefore, this study hypothesizes:

**Hypothesis 3 (H3).** *There is a significant and positive relationship between perceived usefulness and attitude toward online grocery shopping.*

The study by Davis et al. [32] empirically proved the relationship between perceived usefulness and the intention to purchase though online shops. Another study conducted by Gefen et al. [39] also showed the significance of the positive relationship between perceived usefulness and intention to use online platforms for shopping purposes. The

study by Alagoz and Hekimoglu [7] proved that there is also a positive relationship between perceived usefulness and the intention to use e-commerce websites for shopping. Studies on consumer behavior found that the perceived usefulness of an online shop will affect the behavioral intention of customers to purchase online [39,44]. Therefore, this study hypothesizes:

**Hypothesis 4 (H4).** *There is a significant and positive relationship between perceived usefulness and intention toward online grocery shopping.*

### 2.4. Attitude towards Online Grocery Shopping

An attitude toward a certain behavior refers to "the degree to which a person has a favorable or unfavorable evaluation or appraisal of the behavior in question" [47] (p. 188). Moreover, attitude is described in relation to a person's feelings and the tendency towards an object or an idea. Attitude lets people build up a mindset about things liked or disliked. In TAM, the intention of the user towards a new system or technology is strongly affected by the perceived usefulness and their attitude towards using technology [32]. Several studies of online shopping noticed a significant relationship between perceived ease of use, perceived usefulness, and attitude [48–50]. Specifically, Kurnia and Chien [46] investigated numerous factors impacting Australian consumers' acceptance of online grocery shopping and discovered that perceived ease of use and perceived usefulness are the strongest predictors of attitudes. Similarly, in [51], perceived ease of use and perceived usefulness motivated South Korean consumers to use QR codes for food traceability systems. The study by Alagoz and Hekimoglu [7] in the context of emerging countries showed that attitude had a positive correlation with the intention to purchase food products from online platforms. This finding is echoed by Changchit et al. [18], who claimed that attitude exerts a strong impact on Thai consumers' intention to use online grocery shopping. Moreover, Kurnia and Chien [46] asserted that perceived usefulness and attitude have a positive effect on the intention to buy groceries online. Therefore, this study hypothesizes:

**Hypothesis (H5).** *There is a significant and positive relationship between attitude and intention toward online grocery shopping.*

### 2.5. Intention toward Online Grocery Shopping

Behavioral intention is defined as "how hard people are willing to try" and "how much of an effort they are planning to exert" to perform a certain behavior [51] (p. 181). There is a complex process involved when the consumer is deciding to purchase any item. Therefore, purchase intention is always related to a consumer's behavior, attitudes, and perceptions. In TAM, a user's intention regarding using a new system or technology is heavily impacted by the perceived usefulness and attitude [32,52]. Several studies on consumer online shopping have verified the correlations between these constructs [39,53,54]. Quevedo-Silva et al. [55] discovered that attitudes among Brazilian shoppers were positively connected with the desire to purchase products online. Similarly, Loketkrawee and Bhatiasevi [56] found that attitudes had a considerable effect on Thai customers' intent to purchase groceries food online. This is supported by [57], in which there was a significant and positive correlation between attitude and intention to purchase from online shops. Purchase intention is a useful instrument that can forecast the buying process. Drawing upon the perspectives of TAM and prior empirical evidence, this study suggests that behavioral intentions have a positive relationship with actual online shopping behavior. Therefore, this study hypothesizes:

**Hypothesis (H6).** *There is a significant and positive relationship between intention to purchase and actual purchase behavior toward online grocery shopping.*

*2.6. Actual Behavior*

One of the most prominent predictors of actual behavior is intention [58]; hence, analyzing intentions is critical for online stores' success. It has been discovered that online purchase intent is defined as a consumer's willingness to acquire a product via an online platform [59]. In addition to traditional purchases in physical stores, actual buying behavior was examined in various marketing areas such as green marketing [60], luxury brands and products [61], Business to Business (B2B) transactions [62], and, more recently, online shopping [63]. One of the earliest issues of e-commerce development is the lack of intent to shop online [64]. Lim et al. [25] proposed that online purchase intent and actual purchasing behavior should be closely examined.

*2.7. Price*

Price perception refers to a consumer's assessment of a product as appropriately priced, expensive, or inexpensive. The worth of a product and the desire to purchase it are determined by price perceptions. Product pricing is a crucial market decision, and a key success factor for a product, particularly in developing countries, where purchasing decisions are influenced by price [21,65]. Sharma [66] discovered that customers in developing countries are more concerned about price and value than those in developed countries. Rakesh and Khare [67] observed that the vast availability of discounts, rebates, and coupons is one of the primary motivations for emerging market customers to adopt online shopping. Previous research has also found that pricing is a significant factor affecting customers' intention to shop online [68]. Additionally, previous studies showed that price significantly affects online shopping behavior [20,69,70]. Price is one of the least explored areas, particularly with regard to technology adoption [21]. Throughout history, product pricing has been a primary factor influencing a potential buyer's purchasing decisions [71]. In addition, TAM is widely used for understanding the factors affecting consumers' attitudes towards online shopping [72,73], but the measurement of the price dimension is limited. The price factor correlates with the intention to use online shopping and the actual usage. As a result, investigating this potential and understudied variable is necessary to achieve the study's aim, which is to address a gap in knowledge by investigating the influence of price on Indonesian consumers' intent to buy groceries online. Therefore, this study hypothesizes:

**Hypothesis (H7).** *There is a significant and positive relationship between price and intention of online grocery shopping.*

*2.8. Reference Group*

A reference group is a group of people or organization that could affect the behavior of an individual [74]. Regarding online shopping, the reference group relates to perceptions of consumers on the use of online shopping through the opinions of others [36] (p. 434), and proved that comments from a reference group (family, friends, colleagues) positively impact consumer online shopping intention. According to Ha et al. [75], rigidity and social factors have a substantial impact on online buying intentions. Their findings were supported by Kondiparhy et al. [76], who found that, in addition to product comfort, social influences also impacted purchase intention. The same thing was conveyed by other studies [22,23,77] that found a relationship between social influences and purchase intention. However, the relationship between reference group and purchase intention was inconsistent. Several inconsistencies exist between prior studies. According to Bonera [78], the opinions of the reference group do not affect the online purchase intention of online customers. On the other hand, other studies state that the intent of online consumers is positively influenced by the reference group's opinions [34–36]. Therefore, this study hypothesizes:

**Hypothesis (H8).** *There is a significant and positive relationship between the reference group and intention of online grocery shopping.*

*2.9. Health Risk*

The perceived risk theory, in consumer behavior research, refers to the potential risks that may occur in consumers' decision-making, due to uncertainties that may cause negative repercussions [79]. The perceived risk derives from the unanticipated and uncertain negative effects of product purchasing [80]. To date, several studies have recognized the influence of risk perception on life behavior, including consumer behavior. These studies considered risk perception to be an important factor influencing purchasing decisions [81] or purchase intention [82]. Among the different types of perceived risk, health risk indicates that customers perceived a risk to their physical health due to uncontrolled events such as a pandemic [79]. During the current pandemic, virus contagion poses a major threat to society. However, current research has focused on health as a determinant of perceived health risk. The perceived potential and susceptibility of a person to a disease and its severity are terms used to describe perceived health risk [83]. The greater perception a person has of a disease's severity and their susceptibility to it, the more likely they will minimize their chances of contracting the disease [84]. Meanwhile, Salem and Nor [85] noted that, during the pandemic, perceived health risk as a personal sense of potential health hazards is likely to be encountered in physical shopping, e.g., shopping malls and marketplaces. In this pandemic era, several things can be done to prevent the spread of the virus, such as maintaining social distancing, washing hands, and staying at home [86]. Social distancing is the purposeful reduction of close physical contact between individuals in crowded areas such as shopping malls and markets. In decision making, individuals are expected to choose lower-risk choices [87]. Therefore, during the pandemic, the perceived lower risk of using e-commerce compared to other shopping methods may have a positive impact on the intention to engage in online shopping. As a result, someone who perceives a lower risk of using e-commerce will prefer this method to shopping in malls or markets. Therefore, this study hypothesizes:

**Hypothesis (H9).** *There is a positive and significant relationship between perceived health risk and intention of online grocery shopping.*

*2.10. Framework of the Study*

The proposed framework and hypotheses incorporate two TAM constructs (perceived ease of use and perceived usefulness), along with two external factors (reference group and price). Previous research has identified elements that impact purchase intent, including price [69,88] and social influences [22,76]. The authors of [89] conducted a study of TAM variables and added several variables such as awareness, perceived risk, and social influences. The authors of [90] argued that external factors such as confidence perception, risk perception, and other variables were not evaluated as their contributions or effects on TAM were not substantial, although they had an indirect impact on technology acceptance. The study's results support [91], who stated that external factors could be substituted, eliminated, or tailored to the research object and topic. Therefore, Figure 1 proposes the framework as follows.

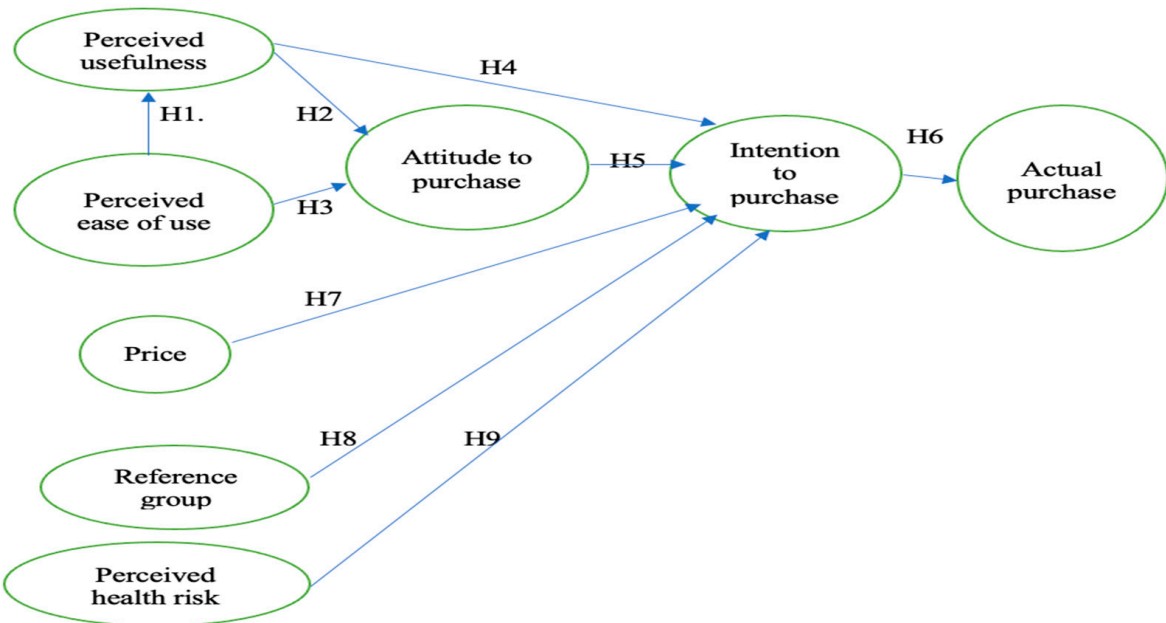

**Figure 1.** Framework of the study.

### 3. Research Methodology

To measure the actual behavior of consumers when purchasing groceries through online platforms in Indonesia, online questionnaires consisting of nine segments (perceived usefulness, perceived ease of use, reference group, price, health risk, attitude, intention, actual behavior, and sociodemographic factors) were distributed to respondents in Jakarta. The study sample was compromised of customers who had recently purchased fruits and vegetables through online platforms. Purposive and snowball sampling methods were applied for this research. The questionnaire was distributed randomly online using the purposive sampling technique, with a focus on respondents who purchased groceries through online shops during the COVID-19 pandemic. The link to the questionnaire was shared through WhatsApp and social media sites such as Facebook and Instagram. Snowball sampling is another technique applied in the current research, in which participants recruit additional participants for a test or study. It is utilized when finding suitable participants is difficult. This sampling approach entails primary data sources recommending additional prospective primary data sources for use in the research. In the current research, a questionnaire was originally distributed to possible respondents, who were requested to complete it and forward it to others. To ensure that participants were qualified, a filtering question was asked before participants were able to access the survey (i.e., Have you purchased groceries online recently?). Before data analysis, data were checked for missing data. No missing data were identified.

Due to the hypotheses involving various assumptions on data distribution, the Partial Least Squares Structural Equation Modeling (PLS-SEM) approach using Smart PLS-3 was used [92]. The sample size used to test the model in PLS was determined by determining the greatest number of formative constructs or the greatest number of antecedents leading to that construct [93]. The minimum sample size was calculated by multiplying the larger value by 10. The model had no formative constructs. The highest numbers of arrows pointing to behavioral purpose was four. Therefore, the minimal sample size was 40, and 150 responses were sufficient to evaluate the data in PLS [92].

The questionnaire was developed based on a previous literature review and modified by the researchers based on a literature search and adjusted with regard to the validity of the content. Except for the section of the sociodemographic items, all other items were measured using a five-point Likert scale, ranging from 1—strongly disagree to 5—strongly agree. The questionnaire development is illustrated in Table 1.

**Table 1.** Questionnaire development.

| Scale | Source |
|---|---|
| **Usefulness** | |
| • Online grocery shopping saves me time.<br>• Using online shopping to purchase groceries makes doing my shopping more efficient.<br>• Online grocery shopping saves me money.<br>• Online grocery shopping helps me to make better purchasing decisions.<br>• Using online shopping to purchase groceries enables me to accomplish transactions more quickly. | [32,42,52] |
| **Ease of use** | |
| • Learning to do online grocery shopping is easy for me.<br>• I find it easy to become skilled at purchasing groceries online.<br>• It is easy to order groceries online.<br>• I think it would be possible for me to shop for groceries online without the help of others. | [36,42,94] |
| **Attitude** | |
| • Purchasing groceries through online shops is a good idea.<br>• Purchasing groceries through online shops is a wise idea.<br>• I like to purchase groceries through online shops.<br>• I think online shopping is beneficial for me. | [36,42,95] |
| **Price** | |
| • The price of a product is a good indicator of its quality.<br>• I don't mind paying more to purchase groceries online.<br>• I have concerns that I may not get my money's worth if I purchase groceries online.<br>• Overall, I am happy with the prices offered for online groceries. | [96–99] |
| **Reference group** | |
| • My best friends and relatives purchase groceries online.<br>• People around me purchase groceries online.<br>• People who influence me think that I should routinely purchase groceries online.<br>• People whose opinions I value prefer that I routinely purchase groceries online. | [98] |
| **Health risk** | |
| • In general, using online shops to purchase groceries involves a lower risk of being infected with COVID-19.<br>• There is a low potential for infection when using online shops during the COVID-19 pandemic.<br>• Shopping in physical stores during the COVID-19 pandemic makes me unsafe. Shopping in physical stores during the COVID-19 pandemic creates a high potential of me being infected with the virus. | [100] |
| **Intention to purchase** | |
| • I intend to use online shops to purchase groceries in the near future.<br>• I predict that I will regularly use online shops for grocery shopping in the future.<br>• I intend to recommend online grocery shopping to others. | [42,101–103] |
| **Actual behavior** | |
| • I often purchase my needed groceries through online shops.<br>• I have been purchasing groceries through online shops on a regular basis. I have been purchasing groceries through online shops for the past six months. | [104,105] |

Data were collected from July to September 2021. Out of 453 questionnaires distributed, 300 were filled in completely and proceeded to further analysis (66.22% response rate). As shown above, the sample size in the current study was well above the required level. The questionnaire was bilingual (English and Bahasa Indonesia).

## 4. Results and Discussion

### 4.1. Profile of the Respondents

The respondent profile showed female customers significantly outnumbering male customers (71% vs. 29%), with the majority in the age range of 40 to 49 years old (40%). This study did not attempt to strike a balance between the gender of respondents and to determine the differences between the genders in terms of their purchasing behavior. This is because the intention of this study was to identify the factors that affect the purchase intention behavior of customers rather than to examine how these behaviors vary as a function of gender. In terms of education level, the majority (60%) had a bachelor's degree and a monthly income of 3.6–10 million rupiah. In terms of occupations, most of the respondents were employees (55.3%). Table 2 presents the profile of the respondents.

**Table 2.** Profile of the respondents.

| Respondents | Number (%) |
|---|---|
| Gender | |
| Male | 87 (29) |
| Female | 213 (71) |
| Age | |
| 18–29 | 41 (13.7) |
| 30–39 | 77 (25.7) |
| 40–49 | 120 (40) |
| >50 | 62 (20.7) |
| Education | |
| High school or below | 9 (3) |
| Diploma | 40 (13.3) |
| Bachelor's | 180 (60) |
| Master's/Ph.D. | 71 (23.7) |
| Level of Income | |
| <3.5 million rupiah | 41 (13.7) |
| 3.6–10 million rupiah | 102 (34) |
| 10–20 million rupiah | 80 (26.7) |
| >20 million rupiah | 77 (25.7) |
| Occupation | |
| Employee | 166 (55.3) |
| Self-employed | 60 (20) |
| Student | 7 (2.3) |
| Housewife | 56 (18.7) |
| Retired | 11 (3.7) |

### 4.2. Reliability and Validity

As recommended, the current study used a two-step approach [106]. First, the measurement model was tested and the structural model was examined to evaluate the reliability and validity, and the ability of the model to predict a certain result. The reliability of the items was tested using Cronbach's alpha and composite reliability (CR). As shown in Table 1, Cronbach's alpha and CR exceeded the recommended threshold of 0.7, as suggested [107,108]. The average variance extracted (AVE) was higher than 0.5, which corroborates the convergent validity [109]. A confirmatory factor analysis was conducted to test the measurement model, resulting in the low factor loading value (<0.7) and two items (price—3 and usefulness—3) being removed. All other items exceeded the recommended value. Discriminant validity was also tested using the criterion suggested by [109], which mentioned that the square root of each construct's AVE should have a greater value than the correlations with other latent constructs. Even when cross-loading is taken into account, the construct's factor-loading indicators should be greater than all other loads, with the factor loading cutoff value being greater than 0.7 [92]. It was observed that all factor loadings were greater than their cross loads, which is a sign of discriminant validity. Tables 3 and 4 display the test results.

**Table 3.** Mean, SD, internal consistencies, and item loadings.

| Construct | Mean | SD | Item | Loading | Cronbach's Alpha | Composite Reliability | (AVE) |
|---|---|---|---|---|---|---|---|
| Actual behavior | 3.82 | 1.03 | Act-1 | 0.92 | 0.89 | 0.93 | 0.81 |
| | | | Act-2 | 0.92 | | | |
| | | | Act-3 | 0.86 | | | |
| Attitude | 4.08 | 0.84 | Att-1 | 0.88 | 0.91 | 0.94 | 0.78 |
| | | | Att-2 | 0.90 | | | |
| | | | Att-3 | 0.88 | | | |
| | | | Att-4 | 0.89 | | | |
| Ease of use | 4.25 | 0.89 | EOU-1 | 0.92 | 0.89 | 0.93 | 0.76 |
| | | | EOU-2 | 0.91 | | | |
| | | | EOU-3 | 0.86 | | | |
| | | | EOU-4 | 0.79 | | | |
| Health risk | 4.24 | 0.93 | HR-1 | 0.79 | 0.82 | 0.88 | 0.65 |
| | | | HR-2 | 0.79 | | | |
| | | | HR-3 | 0.81 | | | |
| | | | HR-4 | 0.82 | | | |
| Intention | 3.90 | 0.96 | Int-1 | 0.92 | 0.92 | 0.95 | 0.86 |
| | | | Int-2 | 0.93 | | | |
| | | | Int-3 | 0.92 | | | |
| Price | 3.59 | 0.98 | Pri-1 | 0.80 | 0.73 | 0.85 | 0.65 |
| | | | Pri-2 | 0.78 | | | |
| | | | Pri-4 | 0.84 | | | |
| Reference | 3.63 | 0.98 | Ref-1 | 0.86 | 0.89 | 0.92 | 0.75 |
| | | | Ref-2 | 0.88 | | | |

**Table 3.** *Cont.*

| Construct | Mean | SD | Item | Loading | Cronbach's Alpha | Composite Reliability | (AVE) |
|---|---|---|---|---|---|---|---|
| | | | Ref-3 | 0.89 | | | |
| | | | Ref-4 | 0.84 | | | |
| Usefulness | 4.05 | 0.91 | Use-1 | 0.82 | 0.82 | 0.88 | 0.65 |
| | | | Use-2 | 0.84 | | | |
| | | | Use-4 | 0.78 | | | |
| | | | Use-5 | 0.79 | | | |

**Table 4.** Discriminant validity using the criteria of Fornell and Larcker.

| | Actual | Attitude | Ease of Use | Health | Intention | Price | Reference | Usefulness |
|---|---|---|---|---|---|---|---|---|
| Actual | 0.90 | | | | | | | |
| Attitude | 0.76 | 0.89 | | | | | | |
| Ease of Use | 0.47 | 0.59 | 0.87 | | | | | |
| Health | 0.54 | 0.61 | 0.47 | 0.80 | | | | |
| Intention | 0.84 | 0.83 | 0.53 | 0.59 | 0.93 | | | |
| Price | 0.55 | 0.62 | 0.44 | 0.44 | 0.56 | 0.81 | | |
| Reference | 0.65 | 0.63 | 0.46 | 0.43 | 0.63 | 0.54 | 0.87 | |
| Usefulness | 0.59 | 0.68 | 0.60 | 0.57 | 0.69 | 0.56 | 0.51 | 0.81 |

*4.3. Structural Model*

The hypothetical relationships were then evaluated in the following phase of the investigation. The proposed model was tested using structural equation modeling (PLS-SEM). PLS-SEM is a type of variance-based structural equation modeling, allowing complete theories and concepts to be tested. According to [110], PLS-SEM can predict and identify key constructs and tests an extension of the existing structural theories. Therefore, SEM was used as a measuring instrument in this study. Hair et al. [110] state that, when employing PLS-SEM, $R^2$ measured with the significance level of the path coefficients is the primary evaluation criterion for the structural model. As a result, $R^2$ should be high in comparison to the research discipline. An $R^2$ score of 2 is regarded as high as this study sought to explain customer behavior [110]. As shown in Table 5, the model allocates 70% of the variance to actual behavior, 52% of variance to attitude, 74% of variance to intention, and 36% of variance to usefulness. All $R^2$ values were statistically significant. Another aspect of the structural model's evaluation is its capacity to anticipate specific behaviors. The Stone–Geisser $Q^2$ is the predominant measurement that is often used [109]. As proposed by [110], cross-validated redundancy measured the predictive relevance of the models. As a result, the latent construct is predictively relevant if the $Q^2$ value for a certain endogenous latent variable is larger than zero. A blindfolding process with a d-value of 7 was used to measure $Q^2$. All endogenous latent variables have a $Q^2$ value greater than zero, and hence have predictive capabilities.

**Table 5.** Evaluation of the structural model.

| Construct | $R^2$ | Adjusted $R^2$ | *p*-Value | $Q^2$ |
|---|---|---|---|---|
| Actual behavior | 0.70 | 0.70 | 0.00 | 0.57 |
| Attitude | 0.52 | 0.51 | 0.00 | 0.40 |
| Intention | 0.74 | 0.74 | 0.00 | 0.62 |
| Usefulness | 0.36 | 0.36 | 0.00 | 0.23 |

Figure 2 depicts the path coefficients and significance levels. PLS-SEM uses bootstrapping to acquire standard errors for hypothesis testing, entailing repeated random sampling with a replacement from the original sample in creating the bootstrap sample [110]. For investigation, bootstrapping was carried out with 5000 subsamples, as described by [110]. All hypotheses tested were positive and significant, except for health and price. First, direct relationships were tested. Table 6 summarizes the analysis results.

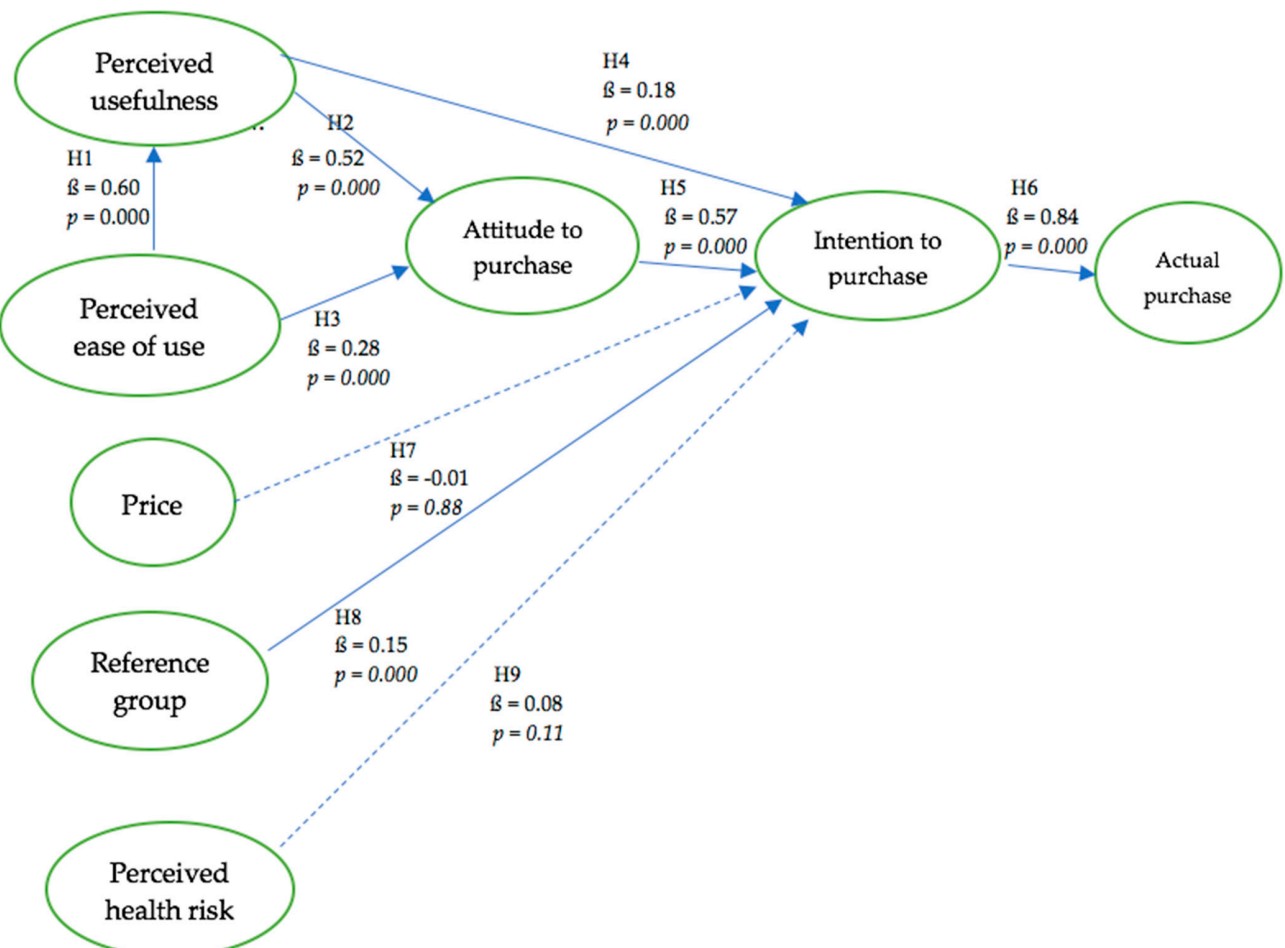

**Figure 2.** Path coefficients of the model.

**Table 6.** Path coefficients.

| Hypothesis | Relationship | ß-Value | T-Statistics | *p*-Value | |
|:---:|:---:|:---:|:---:|:---:|:---:|
| H1 | EOU → Use | 0.60 | 12.33 | 0.00 | Supported |
| H2 | Use → Att | 0.52 | 9.71 | 0.00 | Supported |
| H3 | EOU → Att | 0.28 | 4.77 | 0.00 | Supported |
| H4 | Use → Int | 0.18 | 3.34 | 0.00 | Supported |
| H5 | Att → Int | 0.57 | 10.11 | 0.00 | Supported |
| H6 | Int → Act | 0.84 | 38.92 | 0.00 | Supported |
| H7 | Price → Int | −0.01 | 0.15 | 0.88 | Rejected |
| H8 | Ref → Int | 0.15 | 3.09 | 0.00 | Supported |
| H9 | Health → Int | 0.08 | 1.61 | 0.11 | Rejected |

The hypothesis results are in Table 6, showing the following; EOU → Use (ß = 0.60, t = 12.33, $p < 0.001$), Use → Att (ß = 0.52, t = 9.71, $p < 0.001$), EOU → Att (ß = 0.28, t = 4.77, $p < 0.001$), Use → Int (ß = 0.18, t = 3.34, $p < 0.001$), Att → Int (ß = 0.57, t = 10.11, $p < 0.001$), Int → Act (ß = 0.84, t = 38.92, $p < 0.001$), and Ref → Int (ß = 0.15, t = 3.09, $p < 0.001$). The results show that two hypotheses were rejected: Health → Intention (ß = 0.08, t = 1.61, $p = 0.11$) and Price → Intention (ß = 0.88, t = 0.15, $p = 0.88$).

This study supports the substantial and positive link between ease of use and usefulness (H1), which suggests that grocery shopping is perceived as easy to use and useful. Overall, a mean score of 4.25 was calculated for perceived ease of use, indicating that the majority viewed online grocery shopping platforms as easy to use. In the case of online grocery shopping, the positive association between perceived ease of use and perceived utility was also validated by [46] in Australia [111] in Thailand [112] in Vietnam, and [113] in India.

The results also support the positive relationship between perceived usefulness and buying attitude (H2). The results are comparable to those of other researchers who used other TAM models. Many studies have demonstrated that perceived ease of use has a beneficial effect on customer attitudes [95,114–117]. This positive link may be explained by the online retailer's enhanced efficiency and the effectiveness of the online shopping experience, which gives full information and facilitates product and price comparisons for consumers in their decision making.

A positive and substantial relationship was seen between perceived ease of use and attitudes towards online grocery shopping (H3). This shows that, when customers find technology easy to use, a positive attitude towards it is developed. This conclusion supports prior research that found a substantial association between ease of use and attitudes toward e-commerce [114,116,118,119]. The positive outcome between these two variables might be attributed to the online provider offering high-quality service, including simple and easy-to-use systems such as acceptable, responsive, and informative browsers, order tracking tools, and consumer-accessible after-sales services.

This study also supports the positive association between perceived usefulness and intention to use (H4). The results indicate that usefulness positively influences grocery purchase intention, implying that consumers who find an online food platform useful will use it again in the future. The overall perceived usefulness has a mean value of 4.05, which suggests that online grocery platforms are mostly perceived as useful. A study of behavioral intentions for using a website [120] also found a positive correlation between usefulness and intention to use.

The findings depict a positive relationship between attitude and intention to purchase (H5). The better consumers feel towards an online shop, the more they intend to buy from that online shop. This finding was supported by previous studies [36,95,115,121] that found that consumer attitudes toward using e-commerce would affect their intention to use

e-commerce. Consumers are more inclined to buy groceries online because their behavioral intent is influenced by their attitudes, which influences their actual behavior [43]. As a result, consumers' purchase intention will be influenced by positive thoughts and attitudes.

The positive relationship between purchase intention and actual purchasing (H6) is supported in this study, indicating that consumers who intend to buy groceries online are more likely to do so. The purchase intention had an average value of 3.90. A previous study [122] that focused on business-to-consumer e-commerce found similar results.

A positive relationship between the reference group and intention to purchase (H8) was also supported by this study, contradicting a previous study [78] that did not find a significant correlation between the reference group and purchase intention. A person's sense of social pressure to engage in a behavior is referred to as a reference group. In e-commerce, the reference group represents customer perceptions of the reference group's effect on online purchasing abilities [36]. According to the findings of this study, the opinions of the reference group positively correlate with the purchase intention of online customers. This implies that the reference group promotes online purchasing, resulting in a higher tendency of customers to shop for groceries online.

In contrast, two hypotheses were rejected. The relationship between price and intention to purchase was rejected (H7). There are various possible reasons for that. One possible explanation could be the appropriateness of the prices used for grocery products. Also, it can be concluded that groceries are low-price items. Hence, the benefits of online shopping, which could provide greater savings, are limited. It is possible that more highly perceived products have a greater influence on online shopping intention [123]. Furthermore, in a more realistic context where price differentials cannot be too great for competitive purposes, pricing is unlikely to be as crucial in determining online purchase intention. Finally, the findings revealed that the majority of respondents had a higher level of education and monthly earnings than the typical range of Indonesian wages; therefore, they are less price-sensitive when purchasing groceries online.

In this study, Hypothesis 9 was not supported. In contrast to the prior assumption, a positive, but nonsignificant, link between health and the purchase intent to buy groceries online was found. This result contradicts prior research that found a positive association between health risk and purchasing intention from e-commerce in the COVID era [24,124,125]. Previous studies [126,127] have shown that, during the MERS and SARS pandemics, people with a high education level and a better understanding of the virus perceived a low risk of infection. In comparison to other research, an insignificant association between perceived health risk and intent to shop online was highly probable due to the higher education level of the respondents in this study. The thorough information provided by many credible experts and agencies about COVID-19 risks has helped these educated consumers to develop an understanding of how to avoid infection. Therefore, health risk has no significant impact on the intention to purchase groceries online.

## 5. Conclusions

The coronavirus pandemic has accelerated the adoption of online shopping, particularly for groceries. Online shopping in Indonesia is unlikely to cease or slow down after COVID-19. Indeed, increased online shopping is likely to continue after the pandemic. This study is one of only a few studies about purchasing groceries through online platforms in emerging countries such as Indonesia. The study's findings have effectively demonstrated that TAM is a valid model for understanding online grocery shopping in Indonesia. The results are consistent with previous studies about the use of TAM in online grocery shopping [46,113]. In this investigation, all the TAM model's proposed constructs and correlations were deemed relevant. The findings will aid not just in understanding Indonesian consumers, but also the purchasing behaviors of customers in other Southeast Asian developing countries that believe groceries to be fundamental to their three-meals-a-day eating culture. The COVID-19 situation was an unpredictable push factor for Indonesians to purchase groceries through online platforms. Therefore, since this is still considered

a new behavior, it will be beneficial for marketers and online platforms to know more about customer purchase behavior. Moreover, since Indonesians like to follow the trends of developed countries [3,24], it will be useful to know about the online grocery purchase behaviors of Indonesians. Based on the findings, the participants expressed their willingness to purchase groceries through online platforms.

In terms of the usefulness of purchasing groceries online, the respondents mentioned the convenience and time-saving aspects of these platforms. It is possible to conclude that online platforms must incorporate these aspects into their services and effectively express them in their marketing strategies. When comparing the ease of use and usefulness, usefulness was considered to have a significant impact on customer attitudes about using online platforms. Thus, businesses should emphasize their services' applications and usefulness.

Also, the results supported the relationship between a pressure group and intention to purchase groceries among Indonesians. Companies have noticed that friends and family highly affect individuals' decision making.

### 5.1. Theoretical Implications

This study contributes to the literature on online grocery purchasing behavior in Indonesia by expanding the TAM model. This research will contribute to the study of e-commerce, which enables the introduction of modified TAM technologies for grocery shopping in Indonesia as an emerging economy. The results also provide guidelines for future research that will concentrate on the strengths and remove the weaknesses. This study found that price and health risk do not significantly affect the intention behavior of customers. This finding is new and in contrast with the current literature, which shows the significant effects of those factors on intention behavior in developed countries. Studies conducted in developed countries found that price [14,20,123] and perceived health risk [126–129] were important factors in the decision making of customers regarding purchasing groceries through online shops.

Therefore, the findings add new insights by showing that health risk and price have no significant effect on the online grocery shopping purchase intention of customers.

### 5.2. Practical Implications

This study's results can help marketers and online retailers to improve their marketing strategies and sales performance, particularly during COVID-19. The outcomes may also help online marketers target existing and potential consumers through an effective and efficient e-commerce platform system that offers convenience and lower costs. Online marketers should provide user-friendly and engaging website interfaces so their customers can easily control and understand their purchases.

A knowledge of consumer behavior is essential to understanding their decisions about buying groceries online. Therefore, providing insights for merchants, producers, and scientists could help them to discover important ways to improve e-commerce technologies. The government can benefit from the findings of this study to know better how to motivate people to meet their daily needs through online shopping platforms and therefore reduce physical contact and curb the spread of the virus.

In terms of perceived usefulness, respondents have shown the importance of the time-saving and convenience aspects of online grocery shopping. This suggests that online shops need to incorporate those aspects into their services and effectively communicate these aspects in their marketing campaigns. When comparing the effect of usefulness and ease of use of grocery online shopping, it is important to note that perceived usefulness has a much greater influence on the attitude of customers. Therefore, online shops should prioritize the usefulness of their applications and services. Another important aspect of online grocery shopping is the influence of a reference group on customers' decision making. Online shops need to consider that individuals highly value their friends' and family's opinions. They can benefit from this networking effect and increase their customer base through word-of-mouth marketing.

*5.3. Limitations and Future Suggestions*

There are several limitations to this study. As purposive and snowball sampling were used to choose the sample, caution is needed when generalizing about the results. Consequently, future studies should use probability sampling in order to generalize the findings to the whole population. This study added price, reference group, and health risk as modifications of the TAM model, extended by external variables such as price, reference group, and perceived health risk. As seen in the testing model, two out of three external variables were not supported, which means that further research needs to be conducted to find additional variables that can explain the online grocery shopping behavior of customers. We suggest that future studies could use other variables to learn more about customers' online grocery shopping behavior in emerging markets, e.g., convenience, service quality, accuracy, and reasonable prices. For the purpose of identifying external variables, we also suggest conducting a qualitative study in order to determine unknown variables. The data for this study were collected from Jakarta. It is suggested that future studies collect data from other large cities to gain a better understanding of customers across Indonesia.

**Author Contributions:** Conceptualization, investigation, resources, writing—original draft preparation, writing—review and editing, supervision, project administration, D.L.W.; methodology: software, formal analysis, validation, data curation, visualization, R.B.H. All authors have read and agreed to the published version of the manuscript.

**Funding:** This research received no external funding.

**Informed Consent Statement:** Informed consent was obtained from all subjects involved in the study.

**Data Availability Statement:** The data presented in this study are available on request from the corresponding author.

**Conflicts of Interest:** The authors declare no conflict of interest.

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
