# Peer review of "Factors That Drive Actual Purchasing of Groceries through E-Commerce Platforms during COVID-19 in Indonesia"

_sustainability, doi:10.3390/su14063235_

Round 1
Reviewer 1 Report
The primary goal of this study was to identify the factors influencing consumer attitudes and intentions on online grocery shopping in Indonesia. The study expands the TAM model. The study was well done, and it is easy to read. The methodology applied is appropriate. However, some issues in the paper need to be solved before publication. Here are the suggestions for authors on how to improve their work.
Introduction
Authors should extend the contribution of their work. You say: „This study expands TAM and adds some other variables such as price [20,21], reference group [22,23], and health risk [24], appearing as a crucial factor in online shopping behaviour.
Can you explain more in detail why it is essential to add these variables to contribute to this stream of research?
You claim that „more studies in countries with different cultures are needed than what is currently available in the literature.“
Can you describe why it is essential to carry out such a study in Indonesia from a cultural point of view?
Furthermore, in the introduction section, you should also describe the methodology used in this study.
At the end of the introduction, include one paragraph describing the structure of the paper.
References are needed here:
- „It was found that by using e-commerce to satisfy the needs of the people, the government can achieve its goal to reduce the mobility of residents.“
- „Despite the rapid increase of online shopping in Indonesia based on the above data, there are still limited research on the determinants of consumer attitudes and behaviour related to online grocery shopping.“
- „Another limitation of the previous studies is they focused on final foods or nonperishable products and there is a lack of studies to show about customers purchasing groceries which are perishables through online platforms.“
Literature review
Describe the hypotheses more in detail. Hypotheses H2 and H3 need to be developed separately. For each hypothesis, describe the argument for the hypothesis separately. The same holds for hypotheses H4, H5 and H6.
Research methodology
The authors say that purposive and snowball sampling methods were applied for this research. Can you describe in detail the advantages and disadvantages of this method and why this method was used in your study?
Authors say that the questionnaire was developed based on previous literature reviews and modified by the researcher based on using a literature search and adjusted with regard to the validity of the content.
Provide a table with the items for each construct, and explain what items were modified and how.
How did you handle missing data?
Profile of the Respondents
Profile of the respondents should be part of the methodology, as sample characteristics.
Your sample has more females than men (71% vs 29%).
This indicates that your sample is not representative. Discuss this.
Practical implications
Extend this part. Be more precise in explaining the practical implications of research results.
Theoretical implications should go before practical implications.
Limitations and Future Suggestions
Extend this part related to the representativeness of your study and the snowball sampling method applied.
This study added piece….It is a typo - it should be price.
You say: „It is suggested for future studies to use other variables to know better about customer online grocery shopping behaviour in emerging markets.“
Can you extend this part? Which variables?
Author Response
Thanks for your helpful comments. We added the response in attached file.
Please see the attachment.

Reviewer 2 Report
The paper is well organized. Good introduction and literature review. Adequate methodology and well exposed results. extensive bibliography.
Author Response
Thanks for your time to read our article.
Reviewer 3 Report
This study is a paper dealing with e-commerce in Indonesia, and it deals with what factors make Indonesian consumers purchase food through e-commerce platforms. Although significant outputs are derived from the research results, there are limitations in the following studies, which require faithful supplementation.
- It seems that the title of the paper is insufficient to attract readers' interest. Please change the title to include the contents of the research model.
-
It is questionable whether it is necessary to apply TAM in the research model (Figure 1). Rather, it is judged that it would have been much more meaningful if the model was constructed by deriving the grocery-related attributes that consumers want to purchase through e-commerce.
-
In particular, it would have been better if Indonesian consumers included consumption value-related characteristics that could make them different from people in other countries.
-
Wouldn't it be more appropriate to express it as a reasonable price rather than a price?
-
Please provide detailed survey questions in Table 1.
-
In your description of your data, please be more specific about how you sampled.
-
In Figure 2, if the path coefficient is presented as a standardized beta value rather than a t- value, it will be helpful to understand the influence of the path coefficient.
Author Response
Thanks for your comments and advice.
Please see the attachment.

Reviewer 4 Report
This paper is entitled as Purchasing Groceries through an Online Platform in the Emerging Market of Indonesia in the COVID-19Era. The study was well done, however some issues in the paper need to be solved before publication.
1.Literature review
Describe the hypotheses more in detail
2. Research methodology
In your description of your data, please be more specific about how you sampled.
I recommend that the authors describe in detail the advantages and disadvantages of the Bulgarian snow sealing method and what were the considerations for which you used this method in your study.
3. Practical implications
Explain in more detail the practical implications of the research results
Author Response
Thanks for your helpful comments.
Please see the attachment.

Round 2
Reviewer 1 Report
The authors improved the paper.